# Temporal Dynamics of Oxidative Stress and Inflammation in Bronchopulmonary Dysplasia

**DOI:** 10.3390/ijms251810145

**Published:** 2024-09-21

**Authors:** Michelle Teng, Tzong-Jin Wu, Xigang Jing, Billy W. Day, Kirkwood A. Pritchard, Stephen Naylor, Ru-Jeng Teng

**Affiliations:** 1Department of Pediatrics, Medical College of Wisconsin, Suite C410, Children Corporate Center, 999N 92nd Street, Milwaukee, WI 53226, USA; mteng@mcw.edu (M.T.); twu@mcw.edu (T.-J.W.); xgjing@mcw.edu (X.J.); 2Children’s Research Institute, Medical College of Wisconsin, 8701 W Watertown Plank Rd., Wauwatosa, WI 53226, USA; kpritch@mcw.edu; 3ReNeuroGen LLC, 2160 San Fernando Dr., Elm Grove, WI 53122, USA; billy.day@rngen.com (B.W.D.); snaylor@rngen.com (S.N.); 4Department of Surgery, Medical College of Wisconsin, 8701 Watertown Plank Rd., Milwaukee, WI 53226, USA

**Keywords:** bronchopulmonary dysplasia, oxidative stress, inflammation, endoplasmic reticulum stress, senescence, antioxidant, anti-inflammatory, temporal relationship, therapeutic intervention

## Abstract

Bronchopulmonary dysplasia (BPD) is the most common lung complication of prematurity. Despite extensive research, our understanding of its pathophysiology remains limited, as reflected by the stable prevalence of BPD. Prematurity is the primary risk factor for BPD, with oxidative stress (OS) and inflammation playing significant roles and being closely linked to premature birth. Understanding the interplay and temporal relationship between OS and inflammation is crucial for developing new treatments for BPD. Animal studies suggest that OS and inflammation can exacerbate each other. Clinical trials focusing solely on antioxidants or anti-inflammatory therapies have been unsuccessful. In contrast, vitamin A and caffeine, with antioxidant and anti-inflammatory properties, have shown some efficacy, reducing BPD by about 10%. However, more than one-third of very preterm infants still suffer from BPD. New therapeutic agents are needed. A novel tripeptide, N-acetyl-lysyltyrosylcysteine amide (KYC), is a reversible myeloperoxidase inhibitor and a systems pharmacology agent. It reduces BPD severity by inhibiting MPO, enhancing antioxidative proteins, and alleviating endoplasmic reticulum stress and cellular senescence in a hyperoxia rat model. KYC represents a promising new approach to BPD treatment.

## 1. Introduction

Bronchopulmonary dysplasia (BPD) is the most common lung complication in premature infants [1,2], affecting 50–70% of those born before 28 weeks of gestation [3]. The diagnostic criteria for BPD continue to evolve. The diagnosis of BPD is based on clinical metrics such as the need for oxygen at twenty-eight days of age postnatal and radiographic lung abnormalities [4] or, more often, oxygen requirement beyond the postconceptional age of 36 weeks [5]. Roughly one-fourth to one-third of BPD infants develop pulmonary hypertension [6,7], which is associated with 16% mortality before hospital discharge or 48% mortality within two years of the diagnosis [8]. BPD survivors frequently suffer from recurrent respiratory infections [9], growth delays [10], and reactive airway disease or trouble exercising that can persist into adulthood. The impaired lung growth trajectory in BPD survivors also results in their susceptibility to early-onset chronic obstructive pulmonary disease [11,12].

It has been estimated that more than 15,000 new BPD patients are diagnosed each year in the United States alone [13,14]. Notably, the annual number of newly diagnosed premature BPD neonates in the United States has been unchanged for a considerable period [15]. This intractable situation continues despite various new interventions, including gentle ventilation, judicious oxygen therapy [16], aggressive nutritional support [17], intramuscular vitamin A treatment [18], and early caffeine use [19]. This stagnation underscores the urgent need for further research and more effective interventions. Systemic corticosteroid treatment was once commonly prescribed by clinicians to prevent BPD because of its potent anti-inflammatory properties. After realizing that early systemic corticosteroid treatment, especially within the first week of life, is associated with a significant increase in neurodevelopmental deficit [20], its use has been limited to those with established BPD or after one week of life [21]. Stem cell treatment has been considered a promising treatment for BPD. The benefits of stem cell therapy are derived from its anti-inflammatory and antioxidant activity [22]. Clinical studies have shown that premature neonates tolerate stem cell therapy well, but its benefits for BPD remain to be revealed [22]. Still, concerns regarding possible vascular occlusion, potential tumorigenicity, difficulty obtaining the appropriate number of cells, and maintaining the quality are waiting to be resolved [23]. A meager eight clinical trials investigating therapeutic or supplemental intervention for BPD are listed on Clinicaltrials.gov.

This lack of effective therapies and ongoing clinical trials is because BPD is a complex, multifactorial, and poorly understood disease. A thorough understanding of the pathophysiology and mechanisms of onset and progression, including temporal considerations leading to BPD, is required to provide effective BPD treatments. It should be noted that prematurity and low birth weight are the most critical risk factors for BPD and their association with respiratory distress syndrome (RDS) [1]. Premature rupture of the membranes (PROM), preterm labor, intrauterine growth restriction (IUGR), and maternal hypertension are common causes that lead to premature delivery. Regardless of the cause of preterm delivery, oxidative stress has been reported to be a critical factor in human BPD onset and progression [24].

Similarly, inflammation has been detected in human premature births. Histological evidence of chorioamnionitis is commonly seen in the premature placenta and 94% of peri-viable (21–24 weeks of gestation) placentae [25]. Chorioamnionitis increases OS in the sheep model of premature birth [26]. These observations suggest premature neonates have been exposed to an intrauterine environment with OS and inflammation way before their birth. These observations suggest a more thorough understanding of the complex relationship between OS and inflammation in BPD is needed. 

Premature human neonates born before the alveolar stage do not produce enough surfactant to keep their lungs open, leading to RDS [27]. Supplemental oxygen, positive pressure respiratory support, and exogenous surfactants are usually required for tissue oxygenation. The primary postnatal risk factors of BPD include exposure to elevated levels of OS from mechanical ventilation [28], supplemental oxygen [29], and inflammation [30]. Premature lungs with immature antioxidative systems face an abrupt oxidative stress (OS) challenge beyond their coping capacity at birth. OS activates alveolar macrophages to recruit neutrophils from the circulation as a form of sterile inflammation [31]. Inflammatory cytokines and chemokines released from the inflammatory cells damage lung cells, inhibit angiogenesis and alveologenesis, and encourage tissue fibrosis resulting in BPD. Many other contributors to BPD in premature human neonates, such as hemodynamically significant patent ductus arteriosus, necrotizing enterocolitis, and nosocomial infection [32], make studying the BPD mechanism in human neonates extremely difficult.

Over the past five decades, research has resulted in a few treatments that reduce BPD. Preventing a premature birth is undoubtedly the most ideal way, which, unfortunately, is unsuccessful [33]. Antenatal corticosteroids given to mothers at risk of preterm delivery can reduce the risk of RDS, but their effects on decreasing BPD remain equivocal. Postnatal systemic corticosteroids were once commonly used by clinicians believing the incidence of BPD could be reduced. Unfortunately, the strong association between its use and severe neurologic deficit [34] led the American Academy of Pediatrics to caution against its routine use [35]. Intramuscular vitamin A injection has been shown to reduce BPD [18], but the route of administration has led to limited acceptance by most practices. An attempt to use enteral vitamin A administration, however, failed to show any decrease in moderate to severe BPD [36]. Early caffeine treatment is the most promising strategy, significantly reducing BPD [19] and yielding better long-term neurological outcomes. We must, however, remember that the reduction in BPD was found in the secondary analysis of that study. Early administration of surfactant, permissive hypercarbia or gentle ventilation, early application of continuous positive airway pressure, and aggressive nutritional intervention are other clinical measures that have been widely implemented to decrease BPD without solid clinical evidence. 

In this review, we evaluate the interaction between OS and inflammation during the development of BPD. Initial literature considerations suggest that OS is the primary process in the onset of BPD, but we argue here that this is possibly an overly simplistic interpretation. Understanding the temporal interplay between OS and inflammation is crucial due to this relationship’s complex and poorly understood nature in BPD [37]. Determining whether OS or inflammation alone, independent parallel contributions, or synergistic coupled contributions, especially considering intrauterine factors, remains a significant challenge. This knowledge gap hinders the development of effective therapeutic strategies for BPD. The failure of antioxidant and anti-inflammatory treatments in clinical settings underscores the need for a deeper investigation into their temporal dynamics. Unraveling this relationship could reveal better insights for intervention and potentially lead to more targeted and successful therapies. Thus, comprehending the timing and interaction of OS and inflammation is essential for advancing our understanding of BPD pathophysiology and improving patient outcomes [32]. We suggest that pharmacologic drug candidates with both antioxidant and anti-inflammatory properties appear to afford more promising efficacious treatments for BPD.

## 2. Literature Search 

### 2.1. Eligibility Criteria

We first submitted the outlines to the editorial office and then prepared this review according to the approval outlines. Three authors (M.T., T-J.W., and X.J.) searched PubMed using keywords including premature birth, mechanism, oxidative stress, inflammation, infection, bronchopulmonary dysplasia, or their combination to identify related reports. Both human and animal studies were included. We prioritized our selection criteria to review articles published in journals with high-impact factors. Only articles published in English were eligible for consideration. 

### 2.2. Inclusion Criteria

Two authors (B.W. and K.A.P.J.) first screened the curated articles based on their expertise and article titles. The senior authors (S.N. and R-J.T.) determined which articles should be included after discussion. For treatment effects, only randomized controlled trials were chosen for review. We included human observational studies, human randomized control trials, and animal models for experimental treatments that were important in explaining the crucial mechanisms.

### 2.3. Exclusion Criteria

We excluded articles that were not written in English.

## 3. Relationship between OS and Inflammation in General Pathologies

OS and inflammation are closely intertwined biological processes that often co-exist and can mutually exacerbate each other (Figure 1). However, the temporal relationship of these events can vary depending on the specific disease state and underlying causal mechanism of disease onset. Several signaling pathways have also been reported in the literature that link OS and inflammation together, including Nrf2, Ets-1, Sirt1/p66Shc, Sirt1/PPAR/PGC-1α, nitric oxide synthases, NADPH oxidases (NOX), Fe^2+^, NLRP3/caspase-1/GSDMD, HMGB1/TLR4/MAPKs/NF-κB, and mTOR/TFEB/NF-κB. This review will focus only on the most commonly discussed ones. As this review is mainly about BPD, a more detailed discussion of mechanisms will be presented in Section 4 (Relationship Between OS and Inflammation in BPD).

### 3.1. Inflammation Preceding OS

Infection and autoimmune diseases are two typical conditions in this category. During infection, inflammatory cells (neutrophils and macrophages) infiltrate the site to combat pathogens. The inflammatory cells produce reactive oxygen species (ROS) as part of the defense mechanism, leading to OS. The most representative example is viral respiratory infections [38]. The ROS generated by inflammatory cells includes hypochlorous acid (HOCl) from myeloperoxidase (MPO), superoxide (O_2_^−^) from NOX, xanthine oxidase/dehydrogenase, uncoupled mitochondrial electron transport chains, and uncoupled endothelial nitric oxidase synthase. Superoxide is the primary, most abundant, but not a potent ROS, as it is relatively unreactive with most cell components. Superoxide is converted to other, more potent ROS through several reactions. The presence of free iron (Fe^2+^) converts superoxide to hydroxyl radical (OH•) through the Fenton reaction. Hydroxyl radical is a highly reactive ROS that quickly penetrates the mitochondrion membrane, peroxidizes lipids, damages DNA/RNA, and kills cells. The three superoxide dismutases (Cu, Zn-SOD, Mn-SOD, extracellular SOD) dismutate superoxide to hydrogen peroxide (H_2_O_2_). H_2_O_2_ is a permeable, non-radical ROS that reacts spontaneously with nitrogen or oxygen nucleophiles and π-bonds to damage biomolecules. Another source of hydrogen peroxide generation is monoamine oxidase, mainly found in the central nervous system. Nitric oxide (NO) generated by nitric oxide synthases reacts with superoxide at a diffusion-limited rate (6.7 ± 0.9 × 10^9^/mol^−1^/s^−1^), which is six times faster than the removal of superoxide by the Cu, Zn-SOD, to form peroxynitrite (ONOO^−^). HOCl and ONOO^−^ are potent ROS and reactive nitrogen species (RNS) that react with almost all biomolecules. Tyrosine-containing proteins frequently react with hypochlorous acid and peroxynitrite, yielding chlorotyrosine and nitrotyrosine as stable and characteristic footprints [39].

In autoimmune diseases, chronic inflammation is driven by an overactive immune response, leading to increased ROS production and, hence, OS [40]. The process is more complex than infections, as basal ROS is critical in maintaining the homeostasis of the adaptive immune response, and antioxidant treatments have been shown to increase the mortality rate paradoxically in humans. Initially, it was believed that TNFα released from chronically activated macrophages leads to releasing an oxidoreductase that causes OS. Recent studies suggested ROS may also serve as signaling molecules in many immune cell relationships and functions that prevent the progression of chronic inflammation [41]. This complex relationship between inflammation and OS might explain the antioxidant paradox in humans.

### 3.2. OS Preceding Inflammation

Metabolic disorders [42] and neurodegenerative diseases [43] are two typical conditions in this category. In diabetes and obesity [44], high glucose and fatty acid levels lead to mitochondrial dysfunction with increased ROS production [45]. The ROS-induced activation of transcription factors (NFκB, AP-1, MAP kinase/Mk2, JAK/STAT, PI3K/AKT/mTOR, and HIF1α) upregulates proinflammatory genes (CAMs, MCP-1, TNFα, IL-1, and TGFβ). It triggers the downstream inflammatory cascades in animal studies [46]. Cells cope with OS through multiple pathways, including the unfolded protein response (UPR) [47], autophagy [48], and cellular senescence [49] in an attempt to survive. The UPR, due to excessive accumulation in the endoplasmic reticulum, can activate neutrophils without infection (sterile inflammation) by upregulating cyclooxygenases and mitochondrial dysgenesis in hyperoxia-exposed rat pup lungs [50]. Autophagy, a mechanism that degrades non-essential biomolecules to obtain raw materials for synthesizing essential biomolecules, can promote and regulate inflammation [51]. Excessive autophagy promotes neutrophil-mediated injury by destructing endothelial cell barriers to facilitate neutrophil invasion, encouraging cytokine and ROS production, and inducing UPR. Senescent cells can consistently release inflammatory cytokines by the senescence-associated secretory phenotype (SASP), which perpetuates inflammatory responses until they are removed by phagocytic cells [49]. 

There is an extensive interaction between UPR, autophagy, and cellular senescence [50]. When OS becomes unopposed or protracted, it can lead to cell death and release intracellular molecules, including the high mobility group box-B1 (HMGB1). HMGB1 is a non-histone nuclear protein stabilizing DNA when it stays in the nucleus. When HMGB1 leaks out from the nuclei to the extracellular compartment, it becomes the most potent damage-associated molecular patterns (DMAP) or pathogen-associated molecular patterns (PMAP) molecule (Table 1) that binds the pattern recognition receptors (PRR) (Table 2), triggering more inflammatory reactions. HMGB1 is also seen in SASP [49]. The receptor for advanced glycation end-products (RAGE), Toll-like receptors (TLRs), CXC chemokine receptor type 4 (CXCR4), macrophage antigen-1, syndecan-3, T cell Ig mucin-3, and CD24-Siglec-10 are the main PRRs for HMGB1 [52]. Once HMGB1 binds the PRR, a cascade of signaling pathways is activated to augment (sterile) inflammation. In Alzheimer’s [53] and Parkinson’s diseases [54], OS from mitochondrial dysfunction or accumulation of damaged proteins (UPR) often occurs first, leading to chronic neuronal inflammation. 

### 3.3. Reciprocal Influence between Inflammation and OS

In many diseases, inflammation and OS form a damaging, synergistic cycle [46]. For instance, in chronic inflammatory diseases, persistent ROS production causes tissue damage and perpetuates the inflammatory state. Conversely, ongoing OS can perpetuate inflammation mainly through modulating NFκB signaling [55], leading to the expression of pro-inflammatory cytokines. The temporal relationship between inflammation and OS depends on the nature of the disease and the developmental stage of the animal. One example is neonatal lungs preferentially activate the NFκB pathway upon exposure to hyperoxia compared to adult lungs [56]. These process-specific sequences and interplay can provide insights into disease mechanisms and potential therapeutic targets for any disease state where OS and inflammation are involved in causal onset and progression. 

## 4. Relationship between OS and Inflammation in BPD

Premature human neonates, especially those born as extremely premature before 28 weeks of gestation, are often subject to mechanical ventilation and supplemental oxygen shortly after birth to maintain metabolic homeostasis. The treatment readily leads to conditions favoring OS onset [50] and a mistaken belief that OS is the dominant factor in BPD causality. However, the situation is somewhat more complicated, as outlined here (Figure 2). Human observational studies demonstrate a strong association between increased OS, inflammation, and premature births. Evidence from animal studies suggests these intrauterine changes contribute to BPD development [57]. The underdeveloped antioxidant system and RDS make premature neonates more susceptible to injury caused by OS and infection. The postnatal OS from respiratory treatment and infection undoubtedly aggravates the injury. Inflammation and OS are the most critical postnatal contributors to BPD, and they inhibit vascular endothelial growth factor A signaling with impaired angiogenesis and alveolar formation and, hence, the BPD phenotype [58].

### 4.1. Antenatal Temporal Phase and BPD

Compared to other complex disease conditions, BPD has several unique features, including the probable involvement of antenatal conditions and events. Considering this, the temporal phase is essential to understanding the role of inflammation and OS in BPD onset and progression. In addition, this temporal phase is heavily influenced by the balance between survival and morbidity, early postnatal fluid management, and nutritional management. Lungs also differ from other organs by directly facing the highest oxygen content and mechanical damage from ventilator support. The high expression of RAGE by type I alveolar epithelial cells, a PRR that binds HMGB1 [59], may also explain why lungs are frequently involved in systemic inflammatory response syndrome.

Approximately one-half of all premature births are considered to have no etiology (idiopathic), which is considered by some researchers as genetically mediated [33]. Other premature births can be the result of preterm labor, PROM, IUGR, or maternal hypertensive disorders. Despite the etiology, ample evidence from human studies suggests that inflammation and OS are highly associated with these antenatal causes of premature delivery through measuring markers for inflammation or OS. The combination of OS and inflammation can result in complex and damaging consequences. 

#### 4.1.1. Intrauterine Inflammation

An ample amount of evidence came from human studies. Peripheral neutrophilia [60] and increased OS [61] in pregnant human women are common occurrences during the third trimester. The intrauterine immune milieu is pro-inflammatory during the first trimester and right before labor but anti-inflammatory between these two periods in human studies [62]. The anti-inflammatory status allows the fetus to develop inside the womb without being rejected by the maternal immune system. The pro-inflammatory status during the first trimester is thought to assist implantation and placentation. The pro-inflammatory status right before labor may help the labor process, protect pregnant women against infections, and even facilitate postpartum recovery. The immune profiling of human cord blood, however, showed inconsistent changes in pro-inflammatory mediators with decreased levels of IL-1β, IL-6, IL-17A, IL-8, eotaxin, MIP-1α, and MIP-1β but increased levels of IL-15 and MCP-1 in premature neonates as compared to term neonates. These findings suggest a reduced capacity for pro-inflammatory immune responses in premature infants in response to maternal inflammation [63]. 

Chorioamnionitis is an acute inflammation of the chorion and membranes of the placenta [64]. We must remember that physiological parturition is an inflammatory process (sterile inflammation) [65] with neutrophil infiltration into the placenta and membranes [66]. This neutrophil infiltration into the maternal–fetal interface during parturition is exaggerated during human preterm birth [67]. The exact role of neutrophils in the onset of labor remains an enigma, as their infiltration is not required for preterm birth in the mouse model of infection-induced preterm labor [68,69]. Unfortunately, obtaining placenta tissue for histology before delivery for studies is impossible, preventing us from studying the temporal changes in human placentas. 

Overall, 1–4% of all human births are complicated by chorioamnionitis [70]. The prevalence varies by diagnostic criteria, risk factors, and gestational age. Around 40% of spontaneous human premature births are considered to be infection-related with inflammatory cell infiltration [71]. It results from ascending infection, often polymicrobial [72], and can occur even with intact membranes. IUGR is a frequent cause of elective preterm delivery. Pregnant women with IUGR also have increased TNFα levels in their blood, indicating an inflammatory status [73]. Sustained maternal inflammation is also known to cause IUGR in sheep [74] and human fetuses [75] and sensitizes the neonate to inflammatory disorders, including BPD [76].

Most obstetricians use clinical findings and laboratory tests to diagnose chorioamnionitis without pursuing microbiology or histology. The clinical findings include a fever of at least 102.2 °F (39 °C), increased white blood cell count, uterine tenderness, abdominal pain, foul-smelling vaginal discharge, fetal and maternal tachycardia, and purulent fluid coming from the cervical os [77]. Occasionally, bacterial culture from amniotic fluid, vaginal discharge, amniotic fluid, or urogenital discharge will be ordered to support the diagnosis. Although histological and clinical chorioamnionitis significantly increases the odds of early onset (<72 h of birth) neonatal sepsis, less than 8% of neonates develop culture-positive early onset neonatal septicemia. However, it is essential to point out that more than 20% of neonates born to mothers with chorioamnionitis develop late-onset (≥72 h of birth) or nosocomial neonatal sepsis, indicating maternal chorioamnionitis increases the susceptibility to infections [78]. An extensive meta-analysis out of 158 human studies (including a total of 244,096 premature infants) showed chorioamnionitis is associated with an increased risk of BPD as defined by oxygen-dependence at 28 days (odds ratio 2.32) or postconceptional age of 36 weeks (odds ratio 1.29). Interestingly, the association between chorioamnionitis and BPD is not a consequence of RDS, as the odds of having RDS do not increase with chorioamnionitis (odds ratio 1.1) [79]. In fact, in an animal study, perinatal maternal antibiotic exposure augments the severity of BPD [80]. 

#### 4.1.2. OS during Pregnancy 

The levels of OS in pregnant women vary during their gestation and reach the highest level in the last trimester [61]. The increased OS may be necessary to stimulate cell proliferation, assist the differentiation and invasion of trophoblasts, and promote placentation [81]. It is hypothesized that increased OS may be required to initiate labor-inducing pathways [82]. Although the exact mechanism by which OS initiates labor remains unclear, increased OS levels have been demonstrated in human preterm labor [24]. One piece of evidence came from a study showing maternal blood levels of malondialdehyde (MDA) were higher in those who delivered premature neonates than those who delivered at term. The same study also showed that maternal blood MDA levels were positively correlated with the corresponding cord blood levels at birth [83]. The evidence suggests that an increased OS may promote premature cellular senescence, with senescence-associated (sterile) inflammation and proteolysis predisposing to the PROM [84].

At birth, human premature neonates have higher levels of OS markers than their full-term counterparts [85], such as plasma F2-isoprostane [86], total and hemoglobin-bound MDA [87,88], erythrocyte membrane hydroperoxides [89], 8-hydroxy-2-deoxyguanosine (8-OHdG) [90], and chelatable iron [86]. Some of these markers are negatively correlated with gestational age [85]. Although there is no difference in maternal blood levels of protein carbonyl between term and premature delivery, the cord blood levels are significantly higher in premature neonates than in term neonates [91]. These findings prove that premature human neonates are born with higher endogenous OS. One common cause that results in premature birth is IUGR, which accounts for over 40% of all induced preterm births before 34 weeks of gestation [92]. It can be idiopathic or, more commonly, secondary to maternal hypertension. Regardless of the etiology, they all result in placental insufficiency with prenatal hypoxia and increased generation of ROS [93]. Irrespective of the etiology, increased OS is consistently seen in premature human neonates. 

### 4.2. Postnatal Temporal Phase and BPD

Neonates born prematurely, especially those born before 32 weeks postconceptional age, frequently require respiratory support, including supplemental oxygen and mechanical ventilation. Both relative hyperoxia from supplemental oxygen and mechanical injury from the ventilator generate OS in the immature lungs. Invasive lines are commonly needed to provide adequate fluid and nutritional support or blood work, offering an opportunity for nosocomial infections. Supplemental oxygen, mechanical ventilation, and infection are well-known risk factors for BPD [50]. Necrotizing enterocolitis has recently been identified to increase the odds of BPD through increased inflammation [94].

#### 4.2.1. Postnatally Increased OS

Preterm infants are susceptible to OS due to an imbalance between the oxidant and antioxidant systems [95]. This imbalance results from insufficient antioxidant capacity due to prematurity and increased OS due to respiratory treatment or infections.

Decreased Antioxidant Capacity

Premature human neonates are born with a deficient antioxidant system, including superoxide dismutase (SOD) [96] and catalase [97] activities in the blood, and SOD and cytosolic glutathione peroxidase activities in red blood cells [89]. The levels of non-enzymatic antioxidants are also decreased in premature neonates, such as vitamin E in blood [81] and red blood cells [89], blood vitamin C [98], reduced glutathione in red blood cells [83], blood transferrin [99], and decreased glutathione in tracheal aspirates [100]. These culminate in lower antioxidant capacity and susceptibility to OS-induced injury in premature neonates.

Supplemental oxygen

The uterus is a relatively hypoxic environment. This relative hypoxia is believed to encourage stabilizing the hypoxia-inducible factors that encourage placentation and fetal lung development [101]. The abrupt transition from an intrauterine oxygen tension of 40–50 torr to an ambient oxygen tension above 140 torr (21% O_2_, room air) is challenging for neonatal lungs. Clinicians face the dilemma of choosing between mortality and morbidity when deciding the optimal oxygen concentration to support neonatal tissue oxygenation. High fractional inspired oxygen (FiO_2_) for neonatal resuscitation is known to be associated with increased OS damage in neonates, especially premature neonates [102]. In neonatal animal studies, both high oxygen concentration and mechanical ventilation decrease hypoxia-inducible factors [99], AMP-activated protein kinase [103], mitochondrial function [104], endothelial nitric oxidase synthase activity [105], matrix metalloproteinase-9 expression [106], and vascular endothelial growth factor abundance [107], which result in arrested alveolarization. 

Mechanical ventilation

Although mechanical ventilation is often life-saving for premature neonates, neonatal animal studies have shown that barotrauma and volutrauma from mechanical ventilation can damage premature lungs and contribute to BPD development by causing OS, inflammation, scarring, and tissue destruction [108,109]. Mechanical ventilation can also cause the downregulation of the vascular endothelial growth factor and its receptor, along with the upregulation of endoglin, which contributes to impaired angiogenesis and alveologenesis [110]. The release of inflammatory mediators also contributes to impaired alveolar formation [111].

Infection

Macrophages and neutrophils activated by infections generate ROS and RNS through NADPH-oxidase (NOX), inducible nitric oxide synthase (iNOS), and MPO. ONOO^−^ from inducible nitric oxide synthase [112] and HOCl, from MPO [113], are the two most potent free radicals generated by myeloid cells in the lungs that contribute to the increased OS (see detailed description in Section 3.1).

Consequence of OS to neonatal lungs

Increased OS in human BPD lungs is evidenced by the increased levels of 8-OHdG, a specific biomarker for OS-induced DNA damage, in histology [114]. Animal studies have shown that hyperoxia-induced OS leads to endoplasmic reticulum stress, mitochondrial dysfunction, activation of myeloid activity [115], sterile inflammation [116], autophagy, apoptosis, and cellular senescence [114]. These sequential events form a self-perpetuated destructive cycle, suppressing alveolar formation (Figure 3).

#### 4.2.2. Postnatal Inflammation

Infection

Multiple studies show a strong association between postnatal infection and human BPD [117,118]. New knowledge has recently emerged from animal studies showing how pattern recognition receptors and DAMP, or PAMP, are involved in the infection-mediated disruption of neonatal lung developmental processes such as angiogenesis, extracellular matrix deposition, and alveolar formation [52] (see detailed description in Section 3.2). Activating the endothelial cell TLR by endotoxin or HMGB1 during infection will lead to deviant vascular formation, epithelial cell injury and reprogramming, disruption of extracellular matrix organization, and pro-inflammatory polarization of the myeloid cells in animal studies. These changes are similar to acute lung injury in adults and can be expected to impact the neonatal lung during the first wave of alveolar formation [117,118].

Sterile inflammation

Animal studies show that an excessive or unopposed OS can elicit sterile inflammation without microorganisms; this type of inflammation results from sterile cell death or injury caused by the OS. Several endogenous molecules released during cellular injury are considered DAMPs [52,119] (Table 1) that initiate sterile inflammation by interacting with PRRs (Table 2). HMGB1 is the most extensively studied DAMP, binding mainly to TLR2, TLR4, and RAGE when released extracellularly. After HMGB1 binds to TLR4 and RAGE, inflammatory cytokines’ transcription, translation, and secretion increase through NFκB signaling [120]. Other interesting facts are that HMGB1–RAGE binding will promote TLR4 translocation to the cell surface, and the HMGB1–TLR4 binding will result in increased transcription and translation of RAGE, forming a self-perpetuating vicious cycle [120]. 

ER stress caused by excessive OS will also amplify cytokine-mediated inflammation without pathogens [121] and contribute to cellular senescence [122]. DNA damage by excessive OS will activate a cascade of tumor suppressors that arrest the cell cycle so that DNA repair will succeed [123]. Tumor suppressor TP53 is the master regulator for DNA damage response. TP53 upregulation can result in apoptosis or cellular senescence, which is context-related [124]. Senescent cells can cause chronic inflammation through their characteristic SASP [125]. Inflammatory mediators in SASP, especially HMGB1, can cause chronic sterile inflammation unless phagocytic cells can effectively remove senescent cells.

### 4.3. OS and Inflammation Mutually Affect Each Other in BPD

OS and inflammation are closely related biological processes that can mutually elicit each other [45]. Their complex interdependence might be the root cause of the antioxidant paradox, in which antioxidants fail to protect against OS-mediated human disorders, including BPD [126]. Existing data from human studies show that both OS and inflammation are involved in premature births and BPD. Animal models have not examined the temporal relationship between OS and inflammation in BPD onset and progression. Evidence shows, however, that both HOX and LPS can lead to HMGB1 release, endoplasmic reticulum stress, and cellular senescence in the lungs [50] (Figure 3). There are rare conditions of BPD in which inflammation might precede OS. These conditions are early neonatal pneumonia, necrotizing enterocolitis in late premature or full-term neonates, or postsurgical infections. These rare conditions start from systemic inflammatory response syndrome or neonatal acute lung injury, resulting in BPD after prolonged respiratory support.

Although our extensive literature review and animal experiments cannot reveal the exact temporal relationship between OS and inflammation in BPD, we summarize that they exert a complex interactive role in the onset and progression of BPD. We suggest that additional studies are needed to understand the intrauterine OS and inflammation relationship and that more well-designed animal studies are needed to further clarify this critical temporal relationship.

## 5. Strategies for BPD Treatment

Our current understanding and best practices for BPD can be summarized as follows. It is believed that BPD development starts from RDS due to premature birth. Clinicians might be capable of postponing preterm delivery, but maintaining the pregnancy until term is almost impossible. The antenatal steroid has been proven to decrease the severity of RDS and should be given to women with preterm labor [127]. As mechanical trauma and oxygen-related injury start right after birth, early surfactant replacement therapy, judicious oxygen use, and gentle ventilation strategy should be implemented. Aggressive nutritional support and adequate fluid management will facilitate the healing of injured premature lungs, which must start as early as possible. Stringent hygienic practice that decreases nosocomial infections is also crucial in reducing BPD. Early administration of caffeine and intramuscular vitamin A injection should be prescribed. These best practices have evolved slowly with time in treating BPD, and further cautious developments continue to occur. 

### 5.1. Early Surfactant Replacement

Preventing prematurity and antenatal corticosteroids are the most effective ways to reduce RDS, but obstetricians manage them, and this is beyond the scope of this review. Since premature neonates with RDS require supplemental oxygen and mechanical ventilation, it is hypothesized that early surfactant administration might avoid high oxygen concentrations and prolonged mechanical ventilation and result in reduced BPD. Several new methods of early surfactant administration have been studied. So far, only less invasive surfactant administration (LISA) has shown a significant reduction in BPD for survivors [128]. Currently, two ongoing clinical trials (NCT05711966 and NCT04984057) evaluate the further use of surfactants in BPD treatment (ClinicalTrials.Gov). Results from these two studies may provide helpful information about optimal surfactant treatment strategies for decreasing BPD. 

### 5.2. Judicious Use of Oxygen

#### 5.2.1. During Resuscitation

Randomized studies and meta-analysis in term neonates comparing 100% FiO_2_ and 21% FiO_2_ during resuscitation revealed an unexpectedly decreased survival rate in the FiO_2_ 21% group [129]. The results spurred changes in practice to use lower FiO_2_ in the resuscitation of premature neonates [130]. The most recent network meta-analysis showed, however, that higher FiO_2_ (60–90%) for resuscitation results in significant survival in premature neonates born under 32 weeks of gestation [131]. The new finding will undoubtedly stir up a debate about the optimal FiO_2_ for premature neonates. 

#### 5.2.2. After Initial Stabilization

Unlike most term neonates, who frequently recover from initial respiratory distress within days with their fully developed antioxidant system, premature neonates require more extended oxygen support to maintain tissue oxygenation and, hence, cause excessive OS. The appropriate oxygen treatment strategy after initial resuscitation and stabilization that offers the best survival rate with the lowest OS-induced morbidities has been studied for decades. The optimal FiO_2_ for premature neonates after initial resuscitation is challenging to study due to their limited blood amount. The noninvasive tissue oxygen saturation (SpO2) measurement through a pulse oximeter has been adopted as the most practical tool for clinical studies. Results from the “Supplemental Therapeutic Oxygen for Prethreshold Retinopathy Of Prematurity” (STOP-ROP) study clearly showed that maintaining SpO2 above 96% increased BPD without improving the neurodevelopmental outcome [132]. The most recent systemic review showed no difference in either survival rate or neurodevelopmental outcome at 18 to 24 months between those who maintained SpO2 between 85–89% and 91–95% [133]. Maintaining SpO2 91–95% during early postnatal life is thus recommended by most neonatal units [134]. 

#### 5.2.3. After BPD Is Established

When the diagnosis of BPD is established, oxygen use becomes more liberalized for a very different purpose. Experts suggest maintaining SpO2 at 92–95% to promote growth and prevent pulmonary hypertension development [135]. 

### 5.3. Gentle Ventilation

The trauma inflicted by mechanical ventilation is considered a significant contributor to BPD. Noninvasive respiratory support is slowly being accepted by neonatologists worldwide as the best way to support premature neonates [136]. The so-called noninvasive ventilation includes nasal high-frequency ventilation [137], noninvasive neurally adjusted ventilatory assistance [138], nasal intermittent positive pressure ventilation [139], and nasal continuous positive pressure ventilation [140], which have all been reported. Although nasal continuous positive pressure is the most cost-effective gentile ventilation, we must recognize its 30–80% failure rate and the need for endotracheal intubation [141]. After five large multicenter randomized controlled trials including over 3000 infants born at 24–29 weeks gestation, nearly 50% of surviving infants continue to develop BPD [16]. Although no evidence supports the efficacy, the noninvasive ventilation strategy has become routine for most neonatologists [136]. It is not which ventilatory support is more protective but how aggressive clinicians are willing to de-escalate the aggressive mode of positive airway pressure support that can reduce mechanical trauma to the lungs. 

### 5.4. Antioxidant Treatment for BPD

#### 5.4.1. Selenium 

Selenium is an essential trace element required for the formation of selenoproteins. It also has antioxidant, anti-inflammatory, and anti-aging properties [142]. Although meta-analysis suggested low plasma selenium was associated with increased complications in premature human infants, selenium supplementation did not reduce BPD [143]. 

#### 5.4.2. Superoxide Dismutase (SOD) 

There are three types of SODs, including Cu, Zn-SOD (SOD1), Mn-SOD (SOD2), and extracellular SOD (ec-SOD, SOD3). Animal studies revealed that ec-SOD deficiency aggravates hyperoxia lung injury. At the same time, Mn-SOD overexpression in type II alveolar cells enhances the tolerance to hyperoxia leading to the assumption that SOD treatment might decrease BPD [144]. There are three randomized controlled trials (RCTs) concerning SOD in premature neonates available for meta-analysis. Unfortunately, there is no evidence that SOD reduces BPD [145].

#### 5.4.3. Vitamin C

The encouraging study showing vitamin C treatment in pregnant smokers led to improved lung function in their offspring [146], offering hope for BPD prevention as maternal smoking was considered a risk factor for BPD [50]. Unfortunately, there has been no vitamin C study on premature human infants. High doses of vitamin C and E treatments in premature baboons exposed to prolonged hyperoxia did increase the level of vitamin C and E in blood and tracheal aspirates. Still, they failed to improve lung histology [147].

#### 5.4.4. Vitamin E

Vitamin E is a fat-soluble essential vitamin with eight isoforms [148]. Both α- and γ-tocopherols have anti-inflammatory properties. Maternal α-tocopherol intake during pregnancy is an important growth factor for fetal lung development [149]. The literature has seven randomized controlled trials of high-dose α-tocopherol to prevent BPD. Although one study reported a significant reduction in BPD, the meta-analysis did not show efficacy [150].

#### 5.4.5. N-Acetylcysteine

Glutathione is the largest antioxidant pool in the body. As the precursor of glutathione, *N*-acetylcysteine has been studied in extreme low-birth-weight neonates requiring positive airway pressure support. The multicenter RCT showed no difference in the severity and incidence of BPD by a six-day course of *N*-acetylcysteine started within 36 h of life [151]. However, a recent single-center study showed an encouraging result for antenatal *N*-acetylcysteine treatment in pregnant women showing signs of preterm labor or inflammation that decreased BPD by 90%. This is the only clinical study showing that exogenous antioxidants prevent premature neonates from developing BPD. It deserves our attention that the treatment did not increase the maternal blood glutathione levels. The authors hypothesized that the effect might be epigenetic instead of related to antioxidant activity [152].

#### 5.4.6. Lactoferrin

Lactoferrin is an endogenous protein that has antioxidant activity by binding non-protein-bound iron. Premature neonates who later develop BPD have a more rapid drop of lactoferrin levels in the tracheal fluid within three days [153]. There are three randomized controlled studies using enteral lactoferrin to prevent prematurity-associated morbidities, including BPD. Meta-analysis showed no reduction in BPD by enteral lactoferrin [154].

### 5.5. Anti-Inflammatory Treatment for BPD

System corticosteroids are the only anti-inflammatory agents that have been well-studied in BPD. Although initial human studies showed a reduction in BPD if administered within the first week of life, the early treatment strategy was disappointing. The systemic administration of corticosteroids in the first eight days of life led to a significant increase in long-term neurodevelopmental impairment [34] and other short-term complications such as high blood pressure, high blood sugar, bowel perforation, and bleeding from the stomach or bowel [155]. Systemic corticosteroid treatment after the eighth day might reduce the composite outcome of BPD/death without affecting the neurodevelopmental outcome [156]. When focused on extremely premature neonates, late systemic corticosteroid treatment is beneficial to those with a high risk of death or moderate-to-severe BPD. Still, it may harm those at low risk [21]. It is apparent that clinical experience suggests systemic corticosteroid treatment does decrease BPD but will be associated with an increased risk of neurodevelopmental impairment if started at the onset stage of BPD. 

Non-steroid anti-inflammatory drugs (NSAIDs) have been studied for other purposes. The trial of indomethacin prophylaxis in preterms (TIPP) showed no reduction in BPD [157]. A retrospective cohort study comparing indomethacin prophylaxis within 24 h of birth in extremely premature neonates (<29 weeks or 401–1000 g birth weight) registered in the NICHD Neonatal Research Network database showed no difference in BPD [158]. Another NSAID that is commonly used in premature neonates is ibuprofen. A retrospective study in extremely premature infants (<28 weeks) showed that ibuprofen effectively closed hemodynamically significant patent ductus arteriosus but increased the odds (odds ratio 2.3) of having BPD [159]. Acetaminophen has become a popular NSAID in treating patent ductus arteriosus, but its impact on BPD is presently unknown [160]. 

Although neonatal infection is an independent risk factor for BPD, prophylactic antimicrobial treatment is not recommended due to the high probability of cultivating multi-resistant bacteria. Both early exposure to broad-spectrum antibiotics and prolonged antibiotic treatment in human premature neonates have been shown to increase the odds of moderate to severe BPD [161]. One recent animal study showed that perinatal maternal antibiotic exposure augments the severity of BPD [80]. These findings have been explained by changing the microbiomes in premature neonates, leading to dysbiosis that increases the susceptibility to BPD [162]. Unlike broad-spectrum antibiotics, macrolides are used more specifically to treat Ureaplasma or Mycoplasma infections with anti-inflammatory properties [163]. Among all studied macrolides, only prophylactic azithromycin treatment significantly reduced BPD [164]. There is, however, a lack of pharmacokinetic data and information about the potential side effects of their use in premature neonates.

### 5.6. Pharmacology Agents with Anti-Inflammatory and Antioxidative Properties for BPD

#### 5.6.1. Vitamin A

The efficacy of intramuscular vitamin A injection has been shown to decrease BPD in premature infants [18]. Vitamin A has antioxidative properties due to its hydrophobic chain of polyene units, which can quench singlet oxygen, neutralize thiyl radicals, and stabilize peroxyl radicals [165]. However, we must remember that this antioxidative property is effective only in low-oxygen tension environments. When oxygen tension is high, such as in the lungs under a hyperoxic condition, vitamin A will auto-oxidize. Multiple randomized controlled trials have shown its anti-inflammatory activity [166]. Although systemic analysis demonstrated its efficacy in decreasing BPD [167], clinicians do not commonly use it because it needs to be administered by intramuscular injection [168]. Enteral low-dose vitamin A has been studied in four trials (800 neonates <1500 g birth weight or <32 weeks) but did not show any decrease in BPD by meta-analysis [169]. Inhaled vitamin A offered better protection against BPD than intramuscular injections in the rat hyperoxia model [170]. An NIH-funded Phase-IIB study was recently granted to study the efficacy of inhaled vitamin A on BPD after a successful rat study [171].

#### 5.6.2. Caffeine

Caffeine is a commonly used drug to treat the apnea of prematurity. The successful reduction in BPD in premature neonates from the CAP trial as the secondary outcome analysis [19] prompted us to investigate the mechanisms. We first showed in the rat hyperoxia BPD model that hyperoxia caused endoplasmic reticulum stress, mitochondrial fission, inflammation, and OS-mediated lung injury. Early administration of caffeine effectively reversed these hyperoxia-mediated changes [172]. Our group further showed in the same animal model that caffeine reversed endothelial nitric oxidase synthase uncoupling, at least in part, by preserving the tetrahydrobiopterin levels [112]. The antioxidative effect of caffeine was further demonstrated by Endesfelder et al. by showing decreased H_2_O_2_, malondialdehyde, and 8-OHdG with increased expression of SODs in the hyperoxia rat BPD model [173]. Using an intra-amniotic lipopolysaccharide injection BPD model in rats, Köroğlu et al. demonstrated the anti-inflammatory effect of caffeine with a reduction in the inflammatory response and improved lung morphometry and the measured resistance of the respiratory system [174]. Endesfelder et al. further showed that caffeine achieves its anti-inflammatory effect by decreasing NFkB expression in hyperoxia-exposed neonatal rat lungs [175]. These animal studies show that caffeine attenuates BPD through antioxidative and anti-inflammatory effects. 

#### 5.6.3. Stem Cell Therapy

Stem cell treatment has been considered a promising treatment for BPD. Mesenchymal stem cells (MSC) have attracted the most attention for their ease of isolation, low immune reaction, anti-inflammatory and antioxidative activities [176,177], and reparative properties, mainly in animal studies [22,178]. Studies have shown that MSC therapy reduces inflammation, mitochondrial dysfunction, and fibrosis of neonatal lungs. MSC secretome [179] or exosome [180] are new stem cell therapy modalities that may avoid some of the side effects of MSC therapy. Still, concerns regarding possible vascular occlusion, potential tumorigenicity, and difficulty obtaining the appropriate number of cells and maintaining the quality are waiting to be resolved [23]. A clinical study has shown that human premature neonates tolerate stem cell therapy well, but its benefits for BPD remain to be revealed [22].

#### 5.6.4. N-Acetyl-Lysyltyrosylcysteine Amide (KYC)

KYC is a novel N- and C-capped tripeptide that reversibly inhibits MPO activity to decrease HOCl production [181]. This MPO-inhibiting activity is not seen in vitamin A and caffeine [114]. All reported KYC studies were conducted in animal studies. It has been shown that KYC decreases vascular stress and increases vasorelaxation in sickle cell disease mice [182,183], attenuates experimental autoimmune encephalomyelitis in mice [184], reduces stroke in mice [185], mitigates inflammatory response to rat peritonitis [186], and ameliorates plaque psoriasis in mice [187]. Since neutrophil infiltration is commonly seen in BPD rat lungs shortly after hyperoxia exposure [172], it was hypothesized that KYC might be able to attenuate the severity of BPD by reducing MPO activity. In hyperoxia rat BPD lungs, KYC decreases neutrophil infiltration and MPO expression, indicating anti-inflammatory activity [114]. In that study, the reduced expressions of cyclooxygenases, TLR4, RAGE, and extracellular HMGB1 in BPD lungs further support the presence of an anti-inflammatory property of KYC. It is crucial to know that the anti-inflammatory activity of KYC requires the presence of MPO. MPO transforms KYC into a thiylating species that oxidizes HMGB1 and inhibits HMGB1 binding to TLR4 and RAGE [114]. Reducing ER stress [113], autophagy, and cellular senescence [112] provides more mechanisms by which KYC decreases inflammation in BPD lungs.

The increased protein tyrosine chlorination in BPD rat lungs is significantly decreased by KYC treatment, indicating a reduction in HOCl formation. KYC’s reduced 8-OHdG in BPD rat lungs under immunofluorescent staining agrees with the finding above [114]. KYC facilitates the expression of Nrf2-induced antioxidative enzymes through thiylation and glutathionylation of the Keap1, followed by stabilizing Nrf2, which offers one more mechanism for its antioxidative activity [114]. KYC further reduces ROS formation during the UPR by stabilizing the endoplasmic reticulum [112]. KYC thus has multiple mechanisms to achieve its antioxidative activity.

Compared to vitamin A and caffeine, KYC offers more protection mechanisms against hyperoxia-induced BPD, such as its impact on Nrf2 and HMGB1 signalings. By maintaining a higher count of type 2 alveolar cells for the presumptive resident progenitor cells in neonatal lungs [188] under hyperoxia, KYC might support better lung growth potential. We thus coined the term systems pharmacological agent for KYC to describe its multi-layer protection against BPD. Transcriptomic studies show increased leukocyte (macrophage, neutrophil, and lymphocyte) migration, chemotaxis, degranulation, differentiation, proliferation, and NETosis, with decreased WNT-catenin and Notch signalings in hyperoxia BPD rat lungs. KYC can reverse all these transcriptomic changes in BPD rat lungs. From the transcriptomic results, it is clear that KYC offers more mechanisms to reduce BPD. KYC is currently being studied in an NIH-funded preclinical study for BPD.

## 6. Conclusions

BPD is a common complication of prematurity [1,2]. BPD’s most critical risk factors include supplemental oxygen, mechanical ventilation, and infection [1]. As evidence of increased OS [84] and inflammation [62] are consistently detected in human preterm birth, we believe both prenatal mechanisms play crucial roles in BPD onset. Unfortunately, the clinical reports do not allow us to interpret the temporal relationship between OS and inflammation confidently. The limited clinical studies showing the benefit of antenatal *N*-acetylcysteine in reducing BPD [152] and maternal antibiotic exposure increasing BPD [79] suggest OS is the more critical antenatal mechanism for BPD onset. Postnatal respiratory support and infections undoubtedly add more OS and inflammation to premature lungs and contribute to BPD progression. 

Most neonatologists have already adopted early surfactant replacement and gentle respiratory support in managing premature neonates [189]. However, meta-analysis has yet to confirm the efficacy of gentle respiratory support for BPD. Animal studies have revealed that postnatal OS and inflammation mutually affect each other during BPD development [54], but the available evidence does not allow us to detail the temporal relationship between them. Prolonged use and prophylactic broad-spectrum antibiotics are not recommended for their strong association with the emergence of multi-resistant bacteria and dysbiosis in premature neonates, and surprisingly, a strong association with moderate to severe BPD was reported [161].

Early caffeine treatment and intramuscular vitamin A injection are proven therapeutic strategies that decrease BPD [18,19]. Both reagents have dual antioxidative and anti-inflammatory activities. The original intramuscular vitamin A study improved the survival rate without BPD by 7% compared to the control group [190]. Unfortunately, the three times weekly intramuscular injection to small premature neonates has made it unpopular to most neonatologists [170]. The original CAP trial shows caffeine decreased BPD by 9% [19]. Meta-analysis of five cohort studies showed that early caffeine treatment (<three days) reduces the BPD compared to late (>three days) caffeine treatment but increases the death rate [191]. Another meta-analysis of 15 cohorts shows that high-dose caffeine (>10 mg/kg/d) offers better BPD reduction than low-dose caffeine (<10 mg/kg/d) but with worse weight gain [192]. More studies are thus needed to determine the best starting time and dose of caffeine treatment. 

Although early caffeine and vitamin A treatment reduce BPD by ~10%, more than 35% of premature infants will still develop BPD. There remains room for improvement. As OS and inflammation mutually elicit each other [46], we hypothesize that therapeutic agents with antioxidant and anti-inflammatory properties are needed for BPD treatment. Knowing whether combining intramuscular vitamin A with early caffeine treatment will decrease BPD in premature infants will be essential. The newly developed systems pharmacology agent KYC also holds promise as it involves many layers of protection against BPD. As so many mechanisms are involved in BPD development, a systems pharmacology approach will be required for its treatment.

## Figures and Tables

**Figure 1 ijms-25-10145-f001:**
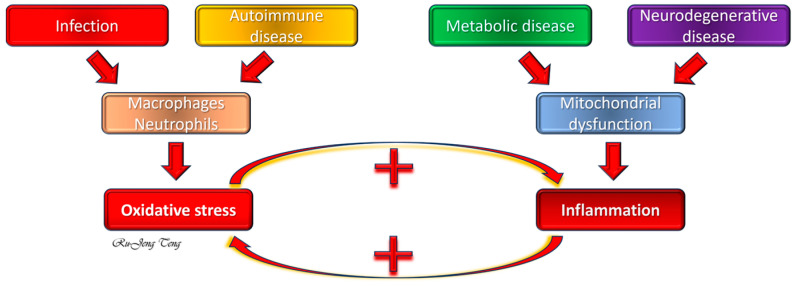
The interaction between oxidative stress (OS) and inflammation in general pathologies. Infection and autoimmune diseases are the most common prototypes that activate macrophages and neutrophils. The reactive oxygen species (ROS) generated by myeloperoxidase (MPO), NADPH oxidase, and uncoupled electron transport chain represent inflammation preceding OS. Metabolic disorders (diabetes mellitus and hyperlipidemia) or neurodegenerative diseases (Parkinson’s disease and Alzheimer’s disease) have mitochondrial dysfunction with increased OS, which then results in secondary inflammation. However, OS and inflammation frequently form a vicious cycle and reciprocally elicit each other. →: from upstream to downstream; +: positive effect.

**Figure 2 ijms-25-10145-f002:**
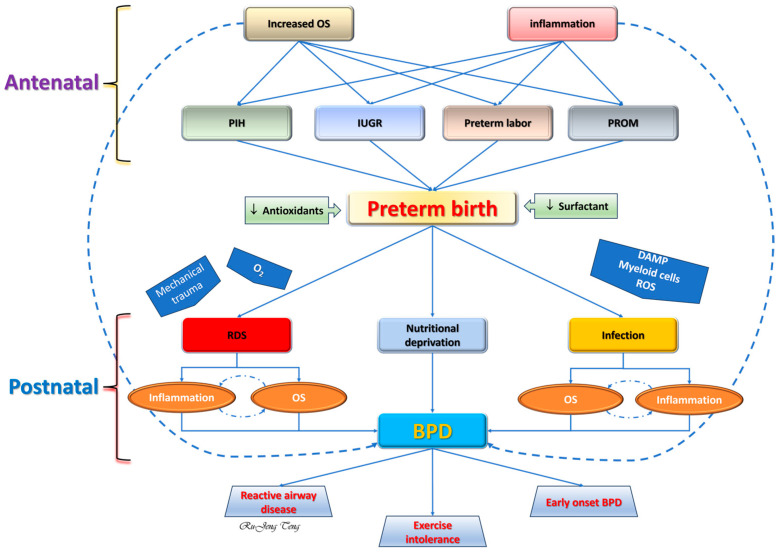
The contributing role of OS and inflammation in developing bronchopulmonary dysplasia (BPD). Increased OS and inflammation are seen antenatally for preterm birth. The surfactant deficiency after preterm birth results in respiratory distress, and antioxidant deficiency leads to sensitivity to OS-induced injury. Postnatal oxygen support, mechanical ventilation, and infection further aggravate OS and inflammation in the premature lung that culminates into BPD. Nutritional deprivation, which weakens the regenerative process, also contributes to BPD. DAMP: damage-associated molecular pattern; IUGR: intrauterine growth restriction; PIH: pregnancy-induced hypertension (pre-eclampsia, eclampsia, and chronic hypertension); PROM: premature rupture of the membranes; ROS: reactive oxygen species. Solid blue arrow: direct relationship; Dashed blue arrow: distant relationship; Broken blue arrow: positive reinforcement.

**Figure 3 ijms-25-10145-f003:**
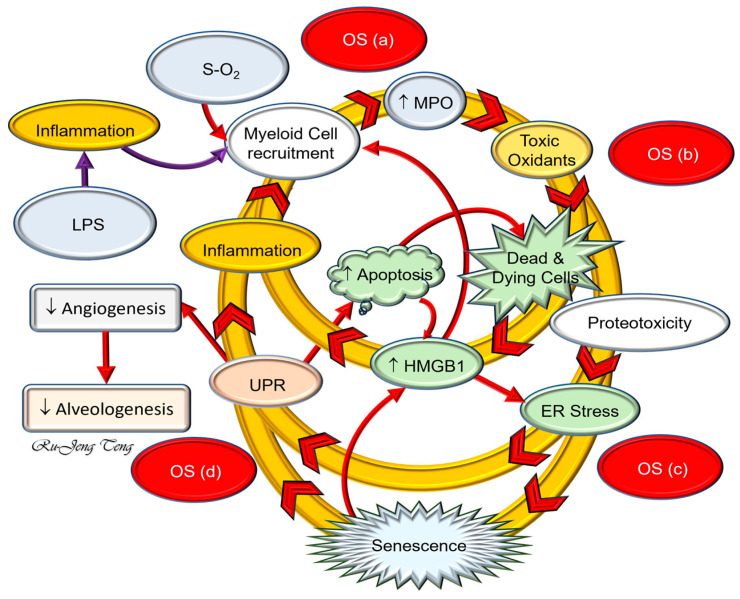
The BPD destructive cycle. The destructive cycle is constructed according to the results of the rat BPD model. OS from hyperoxia or inflammation caused by lipopolysaccharide (LPS) causes myeloid cell infiltration. Myeloperoxidase (MPO) released by macrophages and neutrophils will generate hypochlorous acid (HOCl) from the surrounding chloride anion and H_2_O_2_ released by activated myeloid cells. As one of the most potent free radicals, HOCl can kill or injure nearby cells by releasing high mobility group box 1 (HMGB1), eliciting endoplasmic reticulum (ER) stress, apoptosis, and cellular senescence. HMGB1 is the most potent DAMP, which binds to Toll-like receptor 4 (TLR4) or the receptor for advanced glycation end product (RAGE). ER stress can generate ROS to regain its ability to fold protein and glycosylate nascent proteins. ER stress will activate sterile inflammation and promote cellular senescence. The senescence-associate secretory phenotype (SASP) can lead to chronic inflammation by releasing pro-inflammatory cytokines (IL6, TNFα, HMGB1). The hypermetabolic state of senescent cells and proliferation arrest of the stem/progenitor cells contribute to impaired alveolar formation. These changes form a complex interaction network and self-perpetuate the destructive cycle. The figure is reproduced from [50] under the Creative Commons CC BY 4.0 license). Solid red and purple arrows: direct relationship; Arrowhead: direction of the destructive cycle.

**Table 1 ijms-25-10145-t001:** The damage-associated molecular patterns (DAMPs) molecules. The table is reproduced from [52] under the CC 4.0 license.

Origin	Subcellular Compartment	Molecule
Intracellular	Nuclear	DNA
		Histones
		HMGB1
		HMGN1
		IL1a
		IL33
		RNA
		SAP130
	Cytosol	Aβ
		ATP
		Cyclophilin A
		F-actin
		HSPs
		S100s
		Urate
	Endoplasmic reticulum	Calreticulin
	Mitochondrion	Formyl peptide
		mROS
		mtDNA
		TFAM
	Granule	Defensins
		Cathelicidin
Extracellular matrix		Biglycan
		Decorin
		Fibronectin
		Fibrinogen
		Heparan sulfate
		LMW hyaluronan
		Tenascin C
		Versican

Aβ: amyloid beta; F-actin: filamentous actin; HSP: heat shock protein; HMGB1: high mobility group box 1; HMGN1: high mobility group nucleosome binding domain 1; LMW: low molecular weight; mROS: mitochondrial reactive oxygen species; SAP130: spliceosome-associated protein 130; TFAM: mitochondrial transcription factor A.

**Table 2 ijms-25-10145-t002:** Pattern recognition receptors. The table is reproduced from [52] under the CC 4.0 license.

Family	CDCs	CLRs	FPRs	NLRs	RLRs	TLRs	Scavenger Receptor
Members	AIM2-like receptor	DC-SIGN	FPR1–3	NOD1	LGP2	TLR1–9	CD36
		DEC-2-5		NOD2	MDA5		CD44
		Dectin-1		NLRPs	RIG-1		CD68
		Dectin-2					CD91
		DNGR-1					CXCL16
		Mincle					RAGE
		MMR					
Ligands	DNA	F-actin	Formyl peptide	Aβ	RNA	Biglycan	Calreticulin
		SAP130	Cathelicidin	Biglycan		Decorin	HMGB1
				Histones		DNA	HSPs
				LMW hyaluronan		Fibrinogen	S100s
				mtROS		Glypicans	Versican
				Uric acid		Heparan sulfate	
						Histones	
						HMGB1	
						HSPs	
						LMW hyaluronan	
						mtDNA	
						RNA	
						S100s	
						Syndecans	
						Tenascin C	
						Versican	

AIM2: absent in melanoma 2; CDCs: cytosolic DNA sensor; CLR: C-type lectin receptor; FPR: formyl peptide receptor; LGP2: laboratory of genetics and physiology 2; MDA5: melanoma differentiation-associated protein 5; MMR: mismatch repair; NLR: NOD-like receptor; RIG-I: retinoic acid-inducible gene I.

## Data Availability

The original contributions presented in the study are included in the article; further inquiries can be directed to the corresponding author.

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
