# Peer review of "Temporal Dynamics of Oxidative Stress and Inflammation in Bronchopulmonary Dysplasia"

_ijms, 2024, doi:10.3390/ijms251810145_

Round 1

Reviewer 1 Report

Comments and Suggestions for Authors

Thank you for asking me to review this article aimed to investigatethe role of oxidative stress and inflammation in the development of bronchopulmonary dysplasia (BPD). The issue is relevant and the article is well organizedHowever, I think that the major limit of this work is the lack of a comprehensive analysis of the molecular and biological mechanisms of inflammation and oxidative stress in general pathologies and BPD. By adding this information, the manuscript could increase its strength.

Major comments

The article is well written and every section is clearly reported. Herein, I reported point-to-point comments. 

Abstract: This section is concise and well written. However, there are some spelling and syntax errors that Authors should correct.

Line 32: “Systems pharmacology reagent”. I think that KYC is a pharmacological agent, not reagent. Authors should correct this definition.

Introduction: This section summarizes the definition, epidemiological data and current status of the literature research on BPD. The aim of the study and the goals of the review are well defined. This section is well written.

Main textIn this section, Authors analyzed the relationship between inflammation and oxidative stress in general pathologies and in BPD. However, the molecular and biological mechanisms of inflammation and oxidative stress are not well defined. Authors should better analyze these mechanisms to improve the quality of their discussion. They can refer to the following studies: “Gambadauro, A.; Galletta, F.; Li Pomi, A.; Manti, S.; Piedimonte, G. Immune Response to Respiratory Viral Infections. Int. J. Mol. Sci. 2024, 25, 6178. https://doi.org/10.3390/ijms25116178” AND “” AND “Perrone S, Manti S, Buttarelli L, Petrolini C, Boscarino G, Filonzi L, Gitto E, Esposito SMR, Nonnis Marzano F. Vascular Endothelial Growth Factor as Molecular Target for Bronchopulmonary Dysplasia Prevention in Very Low Birth Weight Infants. Int J Mol Sci. 2023 Feb 1;24(3):2729. doi: 10.3390/ijms24032729.”. Moreover, the paradigm of the interaction between oxidative stress and inflammation is reported in studies on obesity. An example is reported in the following manuscript: “Marseglia, L.; Manti, S.; D’Angelo, G.; Nicotera, A.; Parisi, E.; Di Rosa, G.; Gitto, E.; Arrigo, T. Oxidative Stress in Obesity: A Critical Component in Human Diseases. Int. J. Mol. Sci. 2015, 16, 378-400. https://doi.org/10.3390/ijms16010378”. 

Line 230: “…to cause (IUGR) in sheep…”. Authors can remove the brackets. Line 232-233: Authors should clarify which disease they are referring to (chorioamnionitis?). It is not clear reading the text. Line 272: the acronym “IUGR” was already defined previously. Lines 282-283: “The relative hyperoxia from supplemental oxygen and mechanical injury from the ventilator generates OS in the immature lungs.” Both conditions generate OS or only the second one? Authors should better clarify this sentence. 

Discussion: Authors used this section to summarize topics which they argue previously. However, in this section they repeat concepts already analyzed in the text. They should consider removing this section and reporting the mainly sentences in the “conclusion”.

Line 692: “…from 47% to 36% [19Schmidt].” I suppose the word in the brackets is an error. Authors should modify this sentence. 

ConclusionThis section is concise and well written.

Minor comments 

Throughout the manuscript there are spelling and syntax errors. I suggest that the manuscript is checked by English language editor before final publication. 

Comments on the Quality of English Language

Minor editing is required

Author Response

Comment 1: However, I think that the major limit of this work is the lack of a comprehensive analysis of the molecular and biological mechanisms of inflammation and oxidative stress in general pathologies and BPD. By adding this information, the manuscript could increase its strength.

Response 1: We understand and appreciate the reviewer’s comments. However, there is an extensive list of signaling pathways of inflammation and oxidative stress general pathologies involved.  It is beyond the scope of this review. Still, we have attempted to address the reviewer’s comments and have included signaling pathways relevant to premature births and BPD in the literature. This is shown on pages 4, 4-6, and 8, lines 155-161, 180-240, and 270-275, and highlighted in the enclosed revised draft.  

Comment 2: Abstract: This section is concise and well written. However, there are some spelling and syntax errors that Authors should correct.

Response 2: We appreciate your kind words. We have rewritten the Abstract. As shown in the revised draft.  

Comment 3: Line 32: “Systems pharmacology reagent”. I think that KYC is a pharmacological agent, not reagent. Authors should correct this definition.

Response 3: We agree and have changed it to “agent”.

Comment 4: Introduction: This section summarizes the definition, epidemiological data and current status of the literature research on BPD. The aim of the study and the goals of the review are well defined. This section is well written.

Response 4: Thank you for the encouraging comment.

Comment 5: Main text: In this section, Authors analyzed the relationship between inflammation and oxidative stress in general pathologies and in BPD. However, the molecular and biological mechanisms of inflammation and oxidative stress are not well defined. Authors should better analyze these mechanisms to improve the quality of their discussion. They can refer to the following studies: “Gambadauro, A.; Galletta, F.; Li Pomi, A.; Manti, S.; Piedimonte, G. Immune Response to Respiratory Viral Infections. Int. J. Mol. Sci. 2024, 25, 6178. https://doi.org/10.3390/ijms25116178” AND “” AND “Perrone S, Manti S, Buttarelli L, Petrolini C, Boscarino G, Filonzi L, Gitto E, Esposito SMR, Nonnis Marzano F. Vascular Endothelial Growth Factor as Molecular Target for Bronchopulmonary Dysplasia Prevention in Very Low Birth Weight Infants. Int J Mol Sci. 2023 Feb 1;24(3):2729. doi: 10.3390/ijms24032729.”. Moreover, the paradigm of the interaction between oxidative stress and inflammation is reported in studies on obesity. An example is reported in the following manuscript: “Marseglia, L.; Manti, S.; D’Angelo, G.; Nicotera, A.; Parisi, E.; Di Rosa, G.; Gitto, E.; Arrigo, T. Oxidative Stress in Obesity: A Critical Component in Human Diseases. Int. J. Mol. Sci. 2015, 16, 378-400. https://doi.org/10.3390/ijms16010378”.

Response 5: Thanks for suggesting these three articles for our consideration. We have incorporated your suggestion into the main text (Page 4, lines 180-181; Page 8, lines 275-277; Page 5, Lines 211-212), as well as including the references kindly provided by the reviewer.

Comment 6: Line 230: “…to cause (IUGR) in sheep…”. Authors can remove the brackets.

Response 6: Thanks for pointing out our mistake. We have corrected it.

Comment 7: Line 232-233: Authors should clarify which disease they are referring to (chorioamnionitis?). It is not clear reading the text.

Response 7: Thanks for pointing out this mistake. The word “chorioamnionitis” is added now.

Comment 8: Line 272: the acronym “IUGR” was already defined previously.

Response 8: Thanks for pointing out this mistake. We have removed the spell-out of IUGR.

Comment 9: Lines 282-283: “The relative hyperoxia from supplemental oxygen and mechanical injury from the ventilator generates OS in the immature lungs.” Both conditions generate OS or only the second one? Authors should better clarify this sentence.

Response 9: Both supplemental oxygen and mechanical generate OS in the immature lungs. We have corrected it accordingly (see page 10, line 389 

Comment 10: Discussion: Authors used this section to summarize topics which they argue previously. However, in this section they repeat concepts already analyzed in the text. They should consider removing this section and reporting the mainly sentences in the “conclusion”.

Response 10: We concur with the reviewer’s suggestion. We have condensed and revised the information of the discussion and now added it to the conclusion.

Comment 11: Line 692: “…from 47% to 36% [19Schmidt].” I suppose the word in the brackets is an error. Authors should modify this sentence.

Response 11: Thanks for pointing it out. We have corrected it.

Reviewer 2 Report

Comments and Suggestions for Authors

Dear Authors,

Based on the title I have promised myself a nice review, unfortunately I have found myself, with a need to read extensively long text, which I find difficult to follow. The Tables are too long and are placed in more than one page only. The table footnotes are not clear and the reasons for color differences in the tables are not described. The text has a poor structure without appreciated methodological explanation, based on which the acceptance to include a manuscript has not been described. In the text there is a mixture of data based on animal models and clinical data, which makes the text confusing. I would suggest to restructure the manuscript, include a methodological part explaining criteria on which manuscripts have been accepted. I would suggest to prepare a study flowchart describing inclusion and exclusion criteria. The results should be divided into animal model research and human studies. The discussion is relatively short and does not include necessary parts such as study limitations. The conclusions are also indirectly based on presented data.

Minor comment there is a citation mistake “[19Schmidt]”. In line 694.

Regards,

Comments on the Quality of English Language

long sentences hardly readable

Author Response

Comment 1: Based on the title I have promised myself a nice review, unfortunately I have found myself, with a need to read extensively long text, which I find difficult to follow.

Response 1: We are sorry if the reviewer felt our review did not meet his/her expectations. When we were invited to provide a review article related to our research work, we submitted the outline to the editorial office to ensure that our understanding of the scope met their expectations. Based on editorial responses, we prepared our manuscript according to the approved outlines. The complex interaction between OS and inflammation does not make it easy to interpret the findings straightforwardly, and part of our reasoning for writing this article was to highlight these complexities.

Comment 2: The Tables are too long and are placed in more than one page only.

Response 2: We apologize for not correctly formatting the tables. After manually changing each row’s height, we could fit each table into one page in the revision. We have reformatted the tables so that they can be fitted into the one-page limit.

Comment 3: The table footnotes are not clear and the reasons for color differences in the tables are not described.

Response 3: The table footnotes spell out all abbreviations not described in the main text. The color in Table 1 was intended for better visualization. This appears to cause some confusion, and thus based on this reviewer’s suggestion has now been removed.

Comment 4: The text has a poor structure without appreciated methodological explanation, based on which the acceptance to include a manuscript has not been described.

Response 4: Thanks for the constructive suggestion. However, our manuscript is a qualitative systemic review instead of a quantitative meta-analysis. However, based on the reviewer’s suggestion we have added our strategy of curating articles after searching PubMed. This new text can be found on pages 3-4, lines 132-149.

Comment 5: In the text, there is a mixture of data based on animal models and clinical data, which makes the text confusing.

Response 5: Although animal models have been used to study premature births, it remains controversial whether they faithfully reproduce human premature deliveries. Like most systemic reviews, animal models and clinical data complement each other. Separating the two will make the story fragmented and somewhat more confusing. In our revised manuscript, we address data from humans or animals.

Comment 6: I would suggest to restructure the manuscript, include a methodological part explaining criteria on which manuscripts have been accepted. I would suggest to prepare a study flowchart describing inclusion and exclusion criteria.

Response 6: Thanks for your constructive advice. Due to the simple method of selecting the literature, we think describing how we did it without a flowchart is reasonable.

Comment 7: The results should be divided into animal model research and human studies.

Response 7: We respectfully disagree with the reviewer’s suggestion based on our logic described above.

Comment 8: The discussion is relatively short and does not include necessary parts such as study limitations.

Response 8: We tend to agree with reviewer #1 and his/her suggestions about removing the discussion and incorporating salient points into the conclusion. (see above)

Comment 9: The conclusions are also indirectly based on presented data.

Response 9: We agree that direct evidence is not available. We built our review on both human findings and animal studies. Our goal was/is to review this complex subject matter and draw researchers' attention to this unsettled issue.

Comment 10: Minor comment there is a citation mistake “[19Schmidt]”. In line 694.

Response 10: We apologize for the mistake we made while curating the references. We have made the corrected change.